# Progressive Disease Image Generation with Ordinal-Aware Diffusion Models

**DOI:** 10.3390/diagnostics15202558

**Published:** 2025-10-10

**Authors:** Meryem Mine Kurt, Ümit Mert Çağlar, Alptekin Temizel

**Affiliations:** 1ASELSAN Inc., Ankara 06200, Turkey; 2Graduate School of Informatics, Middle East Technical University, Ankara 06800, Turkey; mecaglar@metu.edu.tr (Ü.M.Ç.); atemizel@metu.edu.tr (A.T.)

**Keywords:** medical image synthesis, diffusion models, Ulcerative Colitis, disease progression modeling, ordinal classification, endoscopy, computer-aided diagnosis, Mayo Endoscopic Score

## Abstract

**Background/Objectives:** Ulcerative Colitis (UC) lacks longitudinal visual data, which limits both disease progression modeling and the effectiveness of computer-aided diagnosis systems. These systems are further constrained by sparse intermediate disease stages and the discrete nature of the Mayo Endoscopic Score (MES). Meanwhile, synthetic image generation has made significant advances. In this paper, we propose novel ordinal embedding architectures for conditional diffusion models to generate realistic UC progression sequences from cross-sectional endoscopic images. **Methods:** By adapting Stable Diffusion v1.4 with two specialized ordinal embeddings (Basic Ordinal Embedder using linear interpolation and Additive Ordinal Embedder modeling cumulative pathological features), our framework converts discrete MES categories into continuous progression representations. **Results:** The Additive Ordinal Embedder outperforms alternatives, achieving superior distributional alignment (CMMD 0.4137, recall 0.6331) and disease consistency comparable to real data (Quadratic Weighted Kappa 0.8425, UMAP Silhouette Score 0.0571). The generated sequences exhibit smooth transitions between severity levels while maintaining anatomical fidelity. **Conclusions:** This work establishes a foundation for transforming static medical datasets into dynamic progression models and demonstrates that ordinal-aware embeddings can effectively capture disease severity relationships, enabling synthesis of underrepresented intermediate stages. These advances support applications in medical education, diagnosis, and synthetic data generation.

## 1. Introduction

Computer-aided diagnosis (CAD) systems have revolutionized endoscopic practice by enhancing the accuracy of lesion detection, standardizing diagnostic criteria, and aiding in clinical decision making [1]. However, developing robust AI models for endoscopic applications remains fundamentally constrained by limited comprehensive datasets that adequately represent the complete spectrum of disease progression. This limitation becomes particularly pronounced when considering the emerging paradigm of personalized disease trajectory prediction, which represents a transformative frontier in medical AI [2,3,4]. Such predictive capabilities, successfully demonstrated across various medical domains through longitudinal symptom monitoring [5,6,7,8], remain largely unexplored in endoscopic applications. This challenge is particularly acute in inflammatory bowel diseases such as Ulcerative Colitis (UC), where current scoring systems inadequately represent the continuous nature of pathological evolution.

Ulcerative Colitis is a chronic inflammatory bowel disease characterized by inflammation and ulceration of the colonic mucosa. The severity of UC is generally assessed using the Mayo Endoscopic Score (MES), which ranges from 0 to 3 [9]; while this scoring system provides a useful standardized clinical assessment, it imposes artificial discrete categories on what is inherently a continuous pathological process [10,11,12]. Although UC severity exists along a continuum, medical image datasets commonly consist of isolated samples annotated with discrete MES levels and controlled image sets capturing the same anatomical region as it progresses through different severity stages are often lacking. In addition, current computer-aided endoscopic systems primarily focus on static image analysis, largely due to this lack of longitudinal data; while the generation of longitudinal disease progression has been explored in other medical domains [8,13,14], there remains a research gap regarding endoscopic applications of disease progression. Such longitudinal data would be valuable for developing comprehensive visual training materials and simulation tools, helping clinicians better recognize subtle transitions between MES stages and gain deeper insight into disease dynamics on a patient-specific level.

Recent advances in generative artificial intelligence, particularly diffusion models, have demonstrated remarkable capabilities in medical image synthesis [15,16]. Unlike Generative Adversarial Networks (GANs), diffusion models offer better training stability, sample diversity, and generation quality, making them particularly suitable for medical applications requiring high accuracy and reliability [17]. On the other hand, since only discrete MES data are available, current generative models can only synthesize discrete UC classes, leaving a significant gap in data generation.

This study addresses these fundamental limitations by introducing novel ordinal class-embedding architectures specifically designed for medical image generation. Our approach converts cross-sectional endoscopic datasets into continuous progression models, enabling the synthesis of realistic intermediate disease stages that are often underrepresented in clinical data. Non-ordinal embedding approaches treat all distinct classes as separate entities in a space. However, for medical datasets, it is imperative to employ ordinal embeddings particularly for the disease severity datasets which are inherently ordinal. The key innovation lies in developing specialized embeddings that capture the cumulative nature of pathological features, recognizing that higher severity levels encompass the characteristics of lower levels while also introducing additional pathological manifestations.

The primary contributions are:Development of ordinality-aware embedding strategies that model disease progression relationships, rather than treating severity levels as independent categories.Adaptation of state-of-the-art diffusion models for medical image synthesis with domain-specific modifications.Generation of realistic synthetic longitudinal datasets to unlock new possibilities for image-based UC trajectory analysis.Exploration of integration with generative data augmentation techniques to improve deep learning model training using synthetic data.

## 2. Related Work

### 2.1. Computer-Aided Diagnosis in Endoscopy

Computer-aided diagnosis has emerged as a transformative technology in endoscopic practice, with deep learning models demonstrating exceptional performance in detecting and classifying gastrointestinal lesions [18]. The Mayo Endoscopic Score (MES) serves as the standard index for evaluating UC disease activity, grading inflammation severity on a scale from 0 to 3: 0 indicates normal mucosa; 1 corresponds to mild disease with erythema and decreased vascular pattern; 2 reflects moderate disease with marked erythema and friability; and 3 denotes severe disease characterized by spontaneous bleeding and ulceration [19].

Existing CAD systems predominantly rely on static image analysis and require extensive datasets covering the full pathological spectrum [20]. Recent advances in edge computing have enabled real-time AI deployment for endoscopic applications [21]; yet, these systems remain focused on static classification tasks, limiting their ability to model temporal disease evolution.

### 2.2. Evolution of Generative Models and Medical Image Synthesis

The progression from traditional generative approaches to modern diffusion models represents a significant advancement in medical image synthesis. The early frameworks included Variational Autoencoders (VAEs) [22] and autoregressive models such as PixelRNN [23], which established foundational probabilistic approaches. Generative Adversarial Networks (GANs) [24] subsequently revolutionized the field through adversarial training between generator and discriminator networks, leading to sophisticated architectures including StyleGAN variants [25,26].

The comparative analysis by Dhariwal and Nichol [27] revealed that diffusion models outperform GANs in image quality and diversity while providing superior training stability. Given these advantages, the research on diffusion-based approaches for medical imaging applications is accelerated, leveraging models’ reliability and consistency.

GAN-based medical image synthesis has been extensively applied across multiple imaging modalities, including radiography, computed tomography, magnetic resonance imaging, and histopathology [17]. Çağlar et al. [28] specifically addressed the classification of Ulcerative Colitis by combining StyleGAN2-ADA with active learning, demonstrating that class-specific synthetic data augmentation enhances classification accuracy when training data are limited, with separate models trained for each Mayo Endoscopic Score category.

Despite these achievements, GANs face substantial limitations in clinical applications, including mode collapse, training instability, and limited sample diversity [17]. These issues are particularly problematic in medical contexts where subtle image features carry significant clinical implications.

Recent diffusion model applications in medical imaging have demonstrated superior capabilities for preserving anatomical details and maintaining clinical validity. Kazerouni et al. [16] provide a comprehensive survey categorizing diffusion models across medical applications including image translation, reconstruction, and generation. Zhang et al. [29] introduced texture-preserving diffusion models for cross-modal synthesis, while Müller-Franzes et al. [17] developed Medfusion, a conditional latent diffusion model specifically designed for medical image generation that outperforms state-of-the-art GANs across multiple medical domains.

### 2.3. Ordinal Relationships in Medical Image Generation

A critical limitation of existing generative approaches is their treatment of disease severity as independent categorical variables rather than ordinal progressions. Takezaki and Uchida [30] introduced an Ordinal Diffusion Model by incorporating ordinal relationship loss functions to maintain severity class relationships during generation. This framework improves distribution estimation through interpolation/extrapolation capabilities, making it particularly applicable to medical conditions with well-defined severity scales.

However, current ordinal approaches primarily focus on discrete classification improvements rather than generating smooth progression sequences that could support medical education and training applications. This represents a significant gap where comprehensive disease progression modeling remains underexplored.

### 2.4. Disease Progression Modeling

Traditional disease progression modeling has relied primarily on longitudinal clinical data and statistical approaches that fail to capture visual manifestations of disease evolution [8]. Recent innovations have explored temporal medical data generation through video synthesis frameworks. Cao et al. [8] introduced Medical Video Generation (MVG), a zero-shot framework for disease progression simulation enabling manipulation of disease-specific characteristics without requiring pre-existing video datasets.

Li et al. [31] developed Endora, combining spatial-temporal video transformers with latent diffusion models for high-resolution endoscopy video synthesis; while these approaches demonstrate promise for temporal content generation, they typically require substantial computational resources and complex training procedures that may limit clinical applicability.

In particular, for UC disease progression, the discrete nature of current scoring systems fails to capture the continuous pathological evolution characteristics [32]. The scarcity of intermediate disease states in clinical datasets restricts the development of comprehensive educational resources and training materials for medical practitioners.

Current approaches to UC image generation have primarily focused on classification tasks rather than progression modeling; while GAN-based methods [28] and ordinal diffusion approaches [30] have shown promise for discrete severity classification, the generation of smooth, clinically meaningful progression sequences remains an unsolved challenge that could significantly benefit medical education and computer-aided diagnosis systems.

## 3. Materials and Methods

This work presents a conditional diffusion framework for synthesizing realistic disease progression sequences from cross-sectional endoscopic data. Conditional diffusion models are selected over alternative generative approaches due to their capability to preserve anatomical details crucial for medical applications [17]. The approach employs specialized ordinal class embeddings that capture the progressive nature of disease severity, enabling the generation of smooth transitions between discrete MES levels while maintaining anatomical consistency. Two novel embedding strategies are introduced: the Basic Ordinal Embedder, which facilitates smooth transitions via linear interpolation between discrete severity classes, and the Additive Ordinal Embedder, which explicitly models the cumulative nature of pathological features. The AOE represents higher disease severity levels as incremental additions to the characteristics of lower levels, thereby capturing the progressive accumulation of pathological changes throughout disease progression.

By adapting the Stable Diffusion v1.4 architecture with medical specific modifications and replacing general purpose text conditioning with domain specific ordinal embeddings, the framework enables precise control over disease severity while maintaining high image quality and training stability.

### 3.1. Model Architecture

The proposed framework adapts Stable Diffusion v1.4 [33] for medical image generation by replacing the standard CLIP text encoder with domain-specific ordinal embeddings. The architecture comprises three key components: (1) a frozen Variational Autoencoder (VAE) for efficient latent space operations, (2) a fine-tuned U-Net backbone for the denoising process, and (3) specialized ordinal class embeddings for medical conditioning. The training pipeline is demonstrated in Figure 1 and the inference pipeline is presented in Figure 2.

The denoising process follows the standard diffusion objective with ordinal class conditioning:(1)$$L=E_{z_{0},\epsilon ,t}\left[||\epsilon -\epsilon_{\theta }\left(z_{t},t,c\right){\left|\right|}^{2}\right]$$
where *c* represents ordinal class conditioning, $z_{t}$ is the noisy latent at timestep *t*, ϵ is added noise, and $\epsilon_{\theta }$ is the predicted noise.

The Basic Ordinal Embedder (BOE) creates learnable embeddings for each integer Mayo Endoscopic Score of 0 to 3 and performs linear interpolation for fractional values:(2)$$E\left(y\right)=\left\{\begin{array}{cc} \left(1-\alpha )E[y]+\alpha E[y+1\right] & \mathrm{if}\;y\notin Z\\ E[y] & \mathrm{if}\;y\in Z \end{array}\right.$$
where y∈[0,K−1] denotes a class label as *K* is the total number of ordinal classes, α=y−⌊y⌋ and $E_{i}\in R^{K\times 768}$ are learnable embeddings.

The Additive Ordinal Embedder (AOE) explicitly models the cumulative nature of pathological features by initializing embeddings monotonically:(3)$$E_{i}=E_{0}+\sum_{j=1}^{i}\Delta_{j},\;i=0,1,2,3$$
where $E_{0}$ represents normal mucosa and $\Delta_{j}\in R^{768}$ represents additive pathological features at severity level *j*.

Compared to the standard CLIP text encoder with approximately 63 million parameters, both proposed ordinal embedders require only 3072 parameters and maintain O(1) inference complexity, demonstrating significantly lower computational complexity.

### 3.2. Dataset and Preprocessing

The study used the Labeled Images for Ulcerative Colitis (LIMUC) dataset [34], comprising 11,276 endoscopic images from 564 patients across 1043 colonoscopy procedures, which is the largest publicly available labeled dataset for UC research. Expert gastroenterologists assigned Mayo Endoscopic Scores (MES) using majority voting for discordant cases. The dataset exhibits significant class imbalance where high severity classes are scare: MES 0 (54.14%), MES 1 (27.70%), MES 2 (11.12%), and MES 3 (7.67%).

Following established protocols [28], 992 images containing medical instruments or annotations were excluded to prevent model bias, resulting in 10,284 images. Images were cropped from 352×288 to 224×224 pixels to remove metadata while preserving the endoscopic view, then resized to 256×256 using bicubic interpolation. Patient-level data splitting ensured no leakage between the training (70%), validation (15%), and test (15%) datasets.

### 3.3. Training Protocol

Training was conducted on NVIDIA V100 GPU, with 16 GB VRAM, batch size 32. The learning rate was set to $1\times 10^{-5}$ for 21,000 optimization steps, corresponding to approximately 672,000 image–conditioning pairs. Three key stabilization strategies were implemented.

Exponential Moving Average (EMA) was applied with a decay rate β=0.999 to reduce training volatility and improve sample quality [35]:(4)$$\theta_{\mathrm{EMA}}\leftarrow \beta \cdot \theta_{\mathrm{EMA}}+\left(1-\beta \right)\cdot \theta$$

Min-SNR-γ Weighting was utilized with γ=1.0 to address gradient conflicts across timesteps and accelerate convergence [36].

Class-Balanced Sampling with inverse class frequencies was implemented to address dataset imbalance. The 30% synthetic data augmentation strategy was implemented to address severe class imbalance, particularly for MES 3, comprising only 7.67% of the dataset. Synthetic images were generated using the GAN-based method from [28], which demonstrated improved minority class representation without compromising model performance. The addition of the synthetic images for the imbalanced dataset can potentially add bias or artifacts from the generative model. However, with an elimination method, such as the core set used in [28], it is possible to inject synthetic samples that respect the inter-class distance and intra-class variety. The 30% proportion was empirically determined to maximize training stability while preventing overfitting to synthetic data patterns.

### 3.4. Evaluation Framework

Four complementary metrics were employed to assess generation quality: Fréchet Inception Distance (FID) [37], CLIP Maximum Mean Discrepancy (CMMD) [38], and precision/recall metrics [39]. This multi-metric approach addresses known limitations of individual metrics and provides comprehensive quality assessment.

Ordinal consistency was assessed using Root Mean Square Error (RMSE), Mean Absolute Error (MAE), classification accuracy, and Quadratic Weighted Kappa (QWK) metrics. Additionally, Uniform Manifold Approximation and Projection (UMAP) [40] analysis compared manifold structures between real and generated datasets using ResNet-18 feature representations.

### 3.5. Experimental Design

Comprehensive comparisons were conducted between proposed ordinal embedding approaches and CLIP text embedder baseline under identical experimental conditions. All methods employed the same diffusion architecture, training procedures, and evaluation protocols to ensure fair comparison. Progressive generation evaluation assessed interpolation capabilities using fine-grained MES increments to examine transition smoothness and anatomical consistency preservation.

Guidance scale of 2.0 provides optimal balance between conditioning strength and natural image appearance. All experiments employed DDPM scheduler for training and inference with 50 denoising steps.

## 4. Results

### 4.1. Quantitative Performance Evaluation

The comparative results across embedding strategies with four complementary image quality metrics are presented in Table 1. AOE achieves better performance in terms of CMMD and Recall metrics, indicating better distributional alignment and improved coverage of the data manifold. Although FID scores are marginally higher for the ordinal methods, they perform better according to the CMMD metric. The literature reports that CMMD provides a more reliable assessment of medical content compared to inception-based metrics [41,42]. The class-wise analysis in Figure 3 reveals that the AOE consistently outperforms alternatives across all Mayo Endoscopic Score severity levels, demonstrating robust performance throughout the entire spectrum of the disease.

The observed non-monotonic performance trends across Mayo Endoscopic Score severity classes in Figure 3 further illuminate the relationship between dataset characteristics and generative model behavior. For less severe classes like MES 0 and 1, the model generates images with increased variety while preserving fidelity, resulting in improved FID scores but reduced precision. Conversely, high-severity classes like MES 2 and 3 suffer from limited training diversity due to their clinical rarity. The generated images for these classes exhibit greater diversity than the constrained training distribution, producing higher FID scores and lower precision while improving recall through enhanced coverage of underrepresented pathological manifestations. This pattern demonstrates that generative models can effectively augment rare disease classes by expanding beyond the limited diversity present in clinical datasets.

### 4.2. Disease Progression Synthesis Results

The framework’s capability to generate clinically meaningful disease progression sequences was evaluated through systematic analysis of synthetic progression trajectories. Figure 4 demonstrates smooth transitions from normal mucosa (MES 0) to severe Ulcerative Colitis (MES 3) in increments of 0.20.

The generated sequences exhibit key characteristics essential for clinical validity, such as maintaining anatomical consistency in the progression stages, the gradual introduction of pathological characteristics corresponding to the Mayo Endoscopic Scoring criteria and realistic intermediate stages. Notably, the framework successfully interpolates between discrete training classes to create plausible intermediate disease states typically underrepresented in clinical datasets.

### 4.3. Validation and Consistency Analysis

Consistency was assessed using a downstream task which is a ResNet-18 regression model trained on the LIMUC dataset to predict continuous MES from generated progression sequences. A total of 1100 progression sequences with 0.1 increments were generated from MES 0.0 to 3.0. Due to the stochastic nature of the diffusion model, which occasionally produces suboptimal images, the best performing 80% of sequences were selected for evaluation, resulting in a total of 27,280 synthetic images. This selection process effectively eliminates stochastic outliers while maintaining sufficient data volume for robust evaluation. A detailed analysis of this process is provided in Appendix B. The Oracle baseline represents the upper bound performance achievable using real test data from the LIMUC dataset, comprising 1443 images, and serves as the reference standard for assessing the validity of synthetically generated progression sequences.

The results shown in Table 2 demonstrate that ordinal embedding approaches achieve ordinal consistency comparable to real data, with Quadratic Weighted Kappa (QWK) scores of 0.8420 and 0.8425 closely matching the test data performance. The CLIP baseline exhibits poor ordinal consistency with QWK of 0.4625, highlighting the critical importance of task-specific embedding approaches for medical severity assessment. More results on various regression models are provided in Appendix A.

Figure 5 illustrates regression predictions for the specific progression sequence shown in Figure 4, demonstrating smooth ordinal relationships maintained across continuous MES values with MAE of 0.45 and RMSE of 0.22.

### 4.4. Ablation Studies and Design Validation

Comprehensive ablation studies confirm the necessity of ordinal aware embedding design. The CLIP baseline fails to maintain clinical consistency across MES levels, producing discontinuous progressions with inadequate interpolation capabilities.

UMAP manifold analysis in Figure 6 demonstrates that generated progression sequences occupy relevant regions of the feature space while providing enhanced coverage of the disease progression continuum. The Silhouette Score of 0.0571 indicates substantial manifold overlap between real and synthetic data, where a score approaching 0 signifies that the manifolds overlap significantly. The synthetic data maintain distributional alignment with real clinical data and extend into intermediate stages, effectively addressing class-imbalance problems often encountered in CAD systems by populating underrepresented classes while preserving the real feature space.

Figure 7 illustrates the fundamental limitation of CLIP embeddings for medical progression modeling. The regression predictions show significant scatter across all MES levels, with predictions failing to follow the expected diagonal progression pattern. This poor ordinal consistency arises because CLIP’s text-based training objective optimizes for semantic similarity between textual descriptions and visual content rather than capturing numerical relationships or ordinal progressions. When MES are converted to text strings like “0.0”, “1.0”, “2.0”, and “3.0”, the embedding space interprets these as distinct semantic categories instead of points along a continuous severity spectrum, preventing meaningful interpolation between severity levels and resulting in discontinuous progression sequences unsuitable for clinical applications requiring smooth disease evolution modeling.

## 5. Discussion

### 5.1. Educational Impact and Applications

The proposed framework shows promise for endoscopic training by generating realistic Ulcerative Colitis progression sequences covering the full disease spectrum. The sequences demonstrate strong ordinal consistency, with a Quadratic Weighted Kappa of 0.84, suggesting suitability for integration into computer-aided diagnosis systems requiring accurate severity assessment. By synthesizing the intermediate disease stages, the framework may help in addressing gaps in medical education where examples of disease progression are limited. Additionally, its ability to produce anatomically consistent sequences with gradual pathological changes supports development of automated assessment and disease trajectory estimation tools for UC severity grading. These results indicate potential to enhance training programs and clinical decision making through synthetic data augmentation of underrepresented disease states.

### 5.2. Technical Contributions and Methodological Advances

The superior performance of ordinal embedding architectures supports the hypothesis that specialized conditioning mechanisms, rather than general-purpose text embeddings, are required for effectively modeling disease progression. The Additive Ordinal Embedder’s design, which models higher severity levels as cumulative accumulations of pathological features, aligns with clinical understanding of UC progression and demonstrates measurable improvements across all evaluation metrics.

Ablation studies reveal that CLIP embeddings fail to capture ordinal relationships essential for progression modeling in medical applications. When MESs are converted to text strings, the embedding space interprets these as distinct semantic categories rather than points along a continuous severity spectrum, which prevents meaningful interpolation between severity levels. This categorical interpretation results in poor clinical consistency, demonstrating why domain-specific embedding design is crucial for medical image generation.

The successful adaptation of diffusion models for medical applications through ordinal conditioning represents a methodological advancement with broader implications. Unlike previous approaches, which have treated disease severity as an independent categorical variable, the framework explicitly models progressive pathological evolution, enabling the generation of clinically meaningful intermediate stages that support both educational and research applications. The proposed approach enables the augmentation of underrepresented classes, particularly those corresponding to higher disease severity levels, by generating synthetic data conditioned on healthy samples. This synthetic data are used to effectively estimate disease progression from the current severity state, allowing simulation of how conditions such as Ulcerative Colitis may worsen or improve over time. Furthermore, our continuous progression framework addresses common issues in previous categorical data generation methods, such as mode collapse and lack of diversity, thereby enhancing the robustness and realism of the generated samples.

### 5.3. Computer-Aided Diagnosis and Clinical Integration

The computational validation results indicate the potential for integrating synthetic progression data into computer-aided diagnosis systems. Generated sequences maintain ordinal relationships which could enhance the robustness of automated assessment systems by providing additional training examples and improving classification accuracy across all severity levels.

For potential real-time endoscopic applications, synthetic progression sequences could serve as reference standards during colonoscopy procedures. This would enable clinicians to compare observed pathology against expected progression patterns. Furthermore, the framework’s ability to generate fine-grained severity increments offers unprecedented granularity for severity assessment when combined with regressive models, potentially supporting more nuanced diagnostic decisions than traditional discrete scoring systems like MES.

UMAP manifold analysis reveals that generated sequences occupy relevant feature space regions while extending coverage to intermediate stages that are underrepresented in clinical datasets. This enhanced coverage is particularly valuable for training robust disease progression models requiring comprehensive pathological spectrum representation, addressing the limitations of datasets that are biased toward discrete MES assessments.

### 5.4. Limitations and Methodological Considerations

Several limitations warrant consideration despite the framework’s demonstrated advantages. The jump in MES 1 shown in Figure 5 indicates that the dataset does not contain smooth transition images from healthy to mild UC, which may suggest that the framework’s reliance on discrete training classes introduces artificial boundaries that do not fully capture the continuous nature of biological processes.

Furthermore, the current approach concentrates on visual progression modeling, omitting temporal dynamics and patient-specific factors that also influence disease evolution. Addressing these limitations in future studies by incorporating longitudinal data and personalized progression modeling approaches would be beneficial; while current computational validation demonstrates promising results, clinical validation by medical professionals is essential before any practical implementations are undertaken in clinical settings.

## 6. Conclusions

This study proposes ordinal class-embedding architectures for conditional diffusion models to generate Ulcerative Colitis progression sequences from cross-sectional endoscopic data. By transforming discrete Mayo Endoscopic Score classifications into continuous progression models, the framework enables the synthesis of intermediate disease stages that are typically underrepresented in traditional datasets. The experimental results show that the Additive Ordinal Embedder performs favorably across metrics.

ResNet-18 regression analysis further suggests that generated sequences preserve medically relevant ordinal relationships and outperform general-purpose CLIP embeddings. This capacity to generate smooth transitions between severity levels addresses limitations of classification-based datasets and offers potential tools for endoscopic training and computer-aided diagnosis. The synthesized intermediate stages could augment training datasets, enhance automated assessment systems, and support clinical decision making across the UC severity spectrum. Additionally, generating more data for underrepresented disease classes may help mitigate data imbalance issues, while the progressive nature of the sequences could facilitate disease trajectory forecasting.

Generating synthetic longitudinal datasets opens new avenues for disease trajectory prediction research. These sequences can enable predictive models to forecast disease evolution from current severity assessments, supporting personalized treatment planning and monitoring. Integration with real-time endoscopic systems offers promising clinical translation potential by enhancing quality assessment and providing intra-procedural feedback. Cross-modal applications linking endoscopic progression with histopathology or clinical biomarkers could further deepen disease understanding and individualized care. Future work should explore advanced data augmentation and interpolation methods to address discrete transitions, standardize evaluation protocols, and develop benchmark datasets to accelerate research and adoption.

Despite these advances, clinical validity must be confirmed by experienced gastrointestinal physicians to ensure practical utility and safety. Future research should also focus on preserving anatomical consistency, capturing temporal dynamics for longitudinal modeling, incorporating additional clinical data, and generalizing the framework to other progressive conditions with ordinal staging.

## Figures and Tables

**Figure 1 diagnostics-15-02558-f001:**
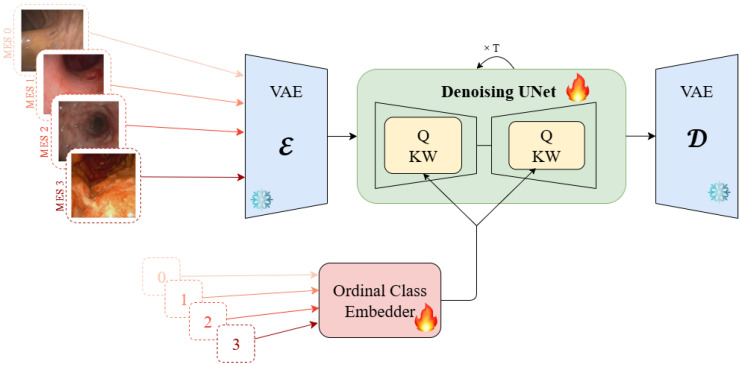
Training pipeline showing the encoding of endoscopic images with Mayo Endoscopic Scores having discrete values as in LIMUC dataset. The denoising U-Net is conditioned on ordinal class embeddings that capture disease severity relationships.

**Figure 2 diagnostics-15-02558-f002:**
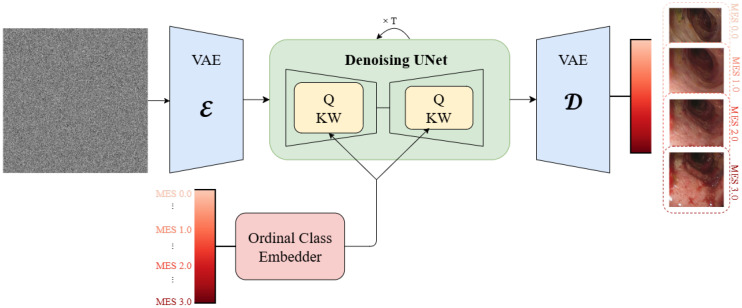
Generation pipeline demonstrating the synthesis process that is able to condition on a continuous values of MES using Ordinal Class Embedder, enabling the generation of intermediate disease stages.

**Figure 3 diagnostics-15-02558-f003:**
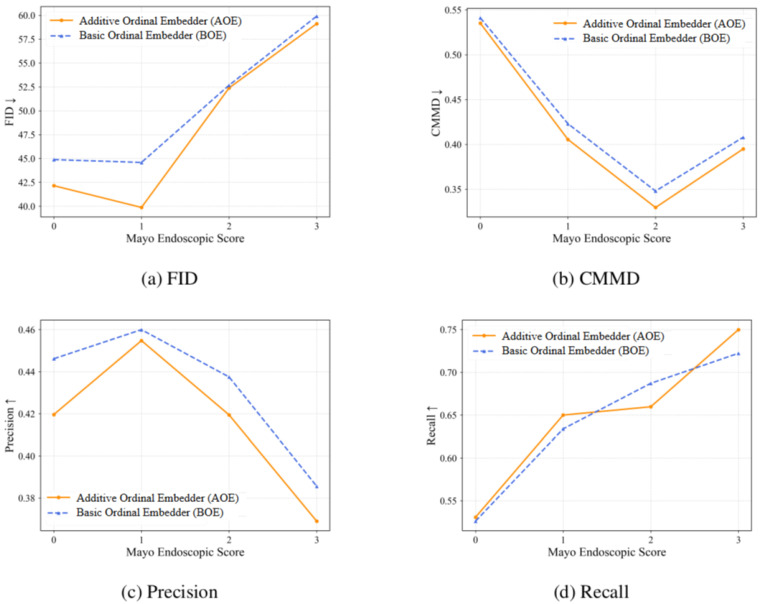
Metric-wise comparison of embedder types across ordinal severity classes. Arrows show whether higher (↑) or lower (↓) values indicate better performance.

**Figure 4 diagnostics-15-02558-f004:**
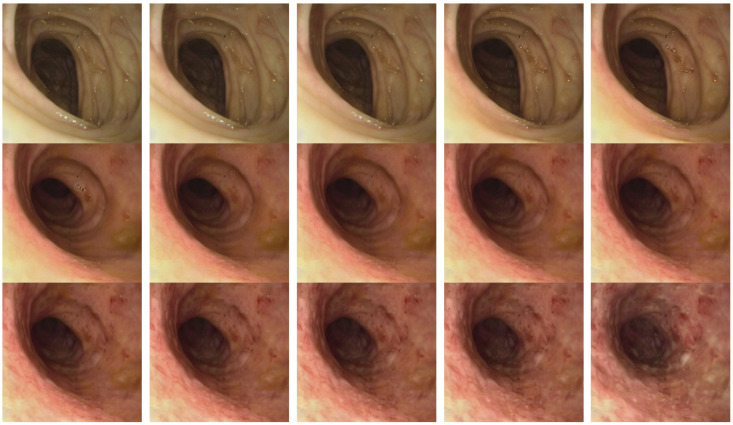
Disease progression sequence generated using the Additive Ordinal Embedder, showing smooth transitions from MES 0 to 3 in increments of 0.20. The progression demonstrates gradual introduction of pathological features while maintaining anatomical consistency.

**Figure 5 diagnostics-15-02558-f005:**
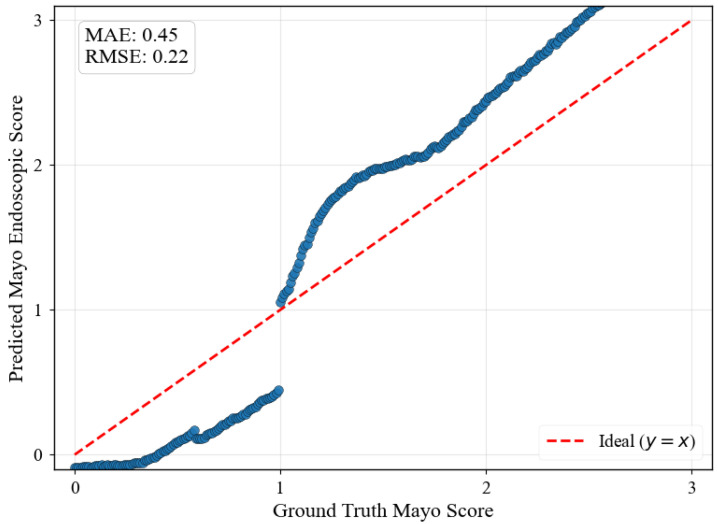
ResNet-18 regression predictions on synthetically generated progression sequences using the Additive Ordinal Embedder with 0.01 increments between MES 0-3. The smooth progression along the ideal diagonal demonstrates successful interpolation between discrete training classes.

**Figure 6 diagnostics-15-02558-f006:**
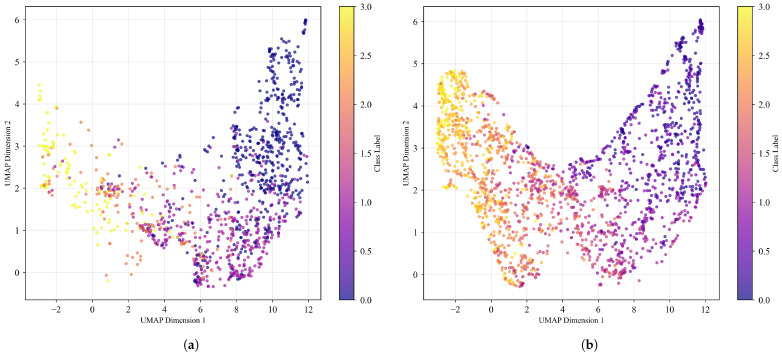
UMAP manifold comparison between real test dataset and generated dataset. The Silhouette Score between these datasets is 0.0571. (**a**) UMAP of LIMUC test dataset; (**b**) UMAP manifold of generated progression dataset by Ordinal Additive Embedder.

**Figure 7 diagnostics-15-02558-f007:**
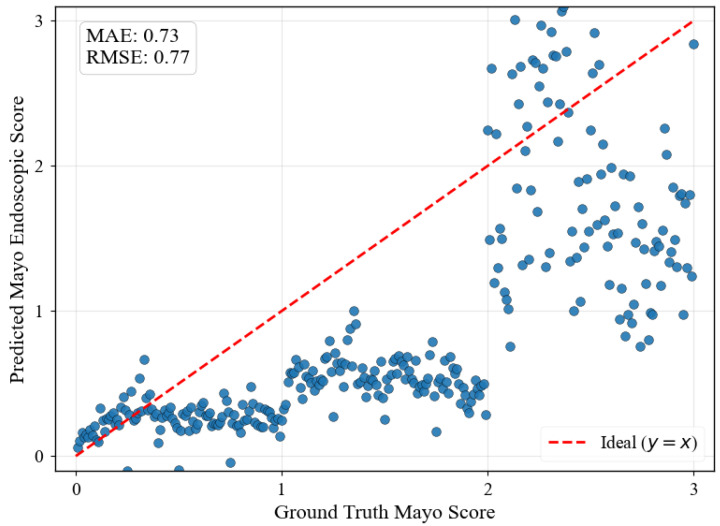
ResNet-18 regression predictions on synthetically generated progression sequences using the CLIP Embedder with 0.01 increments between MES 0-3. The failure in progression indicates that the CLIP Embedder’s embedding space treats classes as distinct semantic categories.

**Table 1 diagnostics-15-02558-t001:** Quantitative comparison of Basic Ordinal Embedder (BOE) and Additive Ordinal Embedder (AOE) methods with the CLIP baseline for Ulcerative Colitis progression generation.

Method	CMMD (↓)	FID (↓)	Precision (↑)	Recall (↑)
CLIP	0.4246	**30.506**	**0.4942**	0.5954
BOE	0.4227	36.072	0.4616	0.6207
AOE	**0.4137**	34.675	0.4614	**0.6331**

Bold values indicate the best performance among the compared methods. Arrows show whether higher (↑) or lower (↓) values indicate better performance.

**Table 2 diagnostics-15-02558-t002:** Ordinal consistency assessment of Basic Ordinal Embedder (BOE) and Additive Ordinal Embedder (AOE) with the CLIP baseline using ResNet-18 regression model.

Dataset	#Images	RMSE	MAE	Accuracy	QWK
Oracle	1443	0.454	0.333	0.7651	0.8591
CLIP	27,280	0.9507	0.7448	0.3896	0.4625
BOE	27,280	0.5171	0.4374	0.6239	0.8420
AOE	27,280	**0.5112**	**0.4238**	**0.6374**	**0.8425**

Bold values indicate the best performance among the compared methods.

## Data Availability

LIMUC (Labeled Images for Ulcerative Colitis) dataset used in this study is publicly available.

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
