# Peer review of "Progressive Disease Image Generation with Ordinal-Aware Diffusion Models"

_diagnostics, 2025, doi:10.3390/diagnostics15202558_

Round 1

Reviewer 1 Report

Comments and Suggestions for Authors

This manuscript introduces a novel ordinal-aware diffusion model, making a valuable contribution to the field of medical image generation. The authors skillfully address the critical challenge of data scarcity by transforming discrete disease scores into continuous and clinically coherent progression sequences.

To further enhance the clarity and impact of the manuscript, I offer the following suggestions and points for consideration:

 Major Suggestions:

1. Clarification on the Filtering of 20% of Sequences: In Section 4.3, the authors state that the best performing 80% of sequences were selected for 
evaluation, resulting in a total of 27,280 synthetic images. This selection process effectively
eliminates stochastic outliers while maintaining sufficient data volume for robust evaluation." This is a potential methodological weakness that could be perceived as "cherry-picking." I recommend the authors:
   - clearly define the criteria for "best performing." Was this based on an automated image quality score, or was it a manual selection process?
   - Provide an analysis of the discarded 20% of sequences. Why did they fail? Did they contain significant artifacts or anatomical inconsistencies? Including examples of failure cases would make the paper more comprehensive and objective.
   - Ideally, the authors should evaluate all generated sequences, report the full results including statistical outliers, and then discuss the distribution and potential causes of the suboptimal images.

 Minor Suggestions:

1. Discussion on GAN data augmentation: In Section 3.3, the authors use a GAN-based method from a previous study for 30% data augmentation to address class imbalance. While this is a practical strategy, a brief comment in the discussion section would be beneficial. Could using GAN-generated data to train a diffusion model potentially introduce biases or artifacts from the GAN itself into the final model? A brief acknowledgment of this potential impact would improve the study's thoroughness.

2. Explanation of the "Gap" in Figure 5: The regression plot in Figure 5 appears to show a small jump or gap around MES 1.0. This is an interesting artifact that warrants a brief comment in the Results or Discussion. Does this reflect a more pronounced visual difference between MES 0 and MES 1 in the training dataset, as alluded to in Section 5.4?

3. Clarification of Interpretation of Figure 3: In Section 5.4, the authors state, "The jump in MES 0 shown in Fig. 3 indicates that..." However, Figure 3 displays performance metrics (FID, CMMD, etc.) across MES classes, not MES values themselves. The authors likely mean there is a significant change in the performance metrics between MES 0 and MES 1. It would be clearer to state this more precisely. For example, for AOE, the FID score improves (decreases) from ~42.5 at MES 0 to ~40.0 at MES 1 before rising again. This non-monotonic trend is worth discussing.

4. Strengthening a Claim with a Citation: In Section 4.1, the authors claim that "CLIP-based metrics provide more reliable assessment than Inception-based metrics for medical content evaluation." This is a strong statement. While plausible, it would be strengthened by citing a reference that supports this view or by briefly explaining the rationale  

Author Response

This manuscript introduces a novel ordinal-aware diffusion model, making a valuable contribution to the field of medical image generation. The authors skillfully address the critical challenge of data scarcity by transforming discrete disease scores into continuous and clinically coherent progression sequences.

To further enhance the clarity and impact of the manuscript, I offer the following suggestions and points for consideration:

 Major Suggestions:

  1. Clarification on the Filtering of 20% of Sequences: In Section 4.3, the authors state that the best performing 80% of sequences were selected for
    evaluation, resulting in a total of 27,280 synthetic images. This selection process effectively
    eliminates stochastic outliers while maintaining sufficient data volume for robust evaluation." This is a potential methodological weakness that could be perceived as "cherry-picking." I recommend the authors:
      - clearly define the criteria for "best performing." Was this based on an automated image quality score, or was it a manual selection process?
      - Provide an analysis of the discarded 20% of sequences. Why did they fail? Did they contain significant artifacts or anatomical inconsistencies? Including examples of failure cases would make the paper more comprehensive and objective.
      - Ideally, the authors should evaluate all generated sequences, report the full results including statistical outliers, and then discuss the distribution and potential causes of the suboptimal images.

We agree with the reviewer that an in-depth discussion of the filtering approach will enhance the article. Accordingly, we have added a new Appendix B, which discusses the criteria for selecting the best-performing synthetic images and provides an analysis of the discarded synthetic samples. The new content in this appendix includes:

  • Appendix B.1 Complete Dataset Performance Assessment: Discussion and comparison of the Additive Ordinal Embedder (AOE) and Basic Ordinal Embedder (BOE).
  • Appendix B.2 Performance-Based Filtering Methodology: Details of the composite score metric combining prediction error and correlation strength.
  • Appendix B.3 Statistical Characterization of Performance Groups: Analysis and comparison of the statistical properties of the 20% discarded subset and the remaining 80% of the data.
  • Appendix B.4 Failure Case Analysis and Clinical Implications: Presents visual samples of common failure cases and discusses the limitations of the generative model failure cases.
  • Appendix B.5 Score Distribution Analysis: Provides a numerical summary of the composite scores for the 20% discarded and 80% high-quality samples..
  • Appendix B.6 Clinical Deployment Considerations: Discusses clinical deployment and future work aspects of our work.

We have defined and grounded performance evaluation according to the performance metric of composite score (Appendix B2). The numerical analysis of this score and the evaluation of the filtered 20% and employed 80% sequences were detailed (Figure A2, Appendix B5). We have also provided an analysis of the discarded sequences, highlighting quantitative discrepancy between the filtered out 20% and high-quality samples (Table A5, Appendix B3). We have also provided discarded samples (Figure A1), possible failure cases (Appendix B4) and discussed clinical deployment considerations (Appendix B6). The complete dataset assessment is also provided for the readership (Appendix B1).

 Minor Suggestions:

  1. Discussion on GAN data augmentation: In Section 3.3, the authors use a GAN-based method from a previous study for 30% data augmentation to address class imbalance. While this is a practical strategy, a brief comment in the discussion section would be beneficial. Could using GAN-generated data to train a diffusion model potentially introduce biases or artifacts from the GAN itself into the final model? A brief acknowledgment of this potential impact would improve the study's thoroughness.

GAN-based generative data augmentation enables generating an extensive synthetic dataset, which is often riddled with duplicates of the original or synthetic images. However, by using an elimination method such as the coreset elimination, it is possible to obtain rare, non-duplicate synthetic images which preserve the inter-class distance and intra-class variety. To briefly acknowledge this potential impact, we have clearly explained this potential in the respective section 3.3, Class-Balanced Sampling.

  1. Explanation of the "Gap" in Figure 5: The regression plot in Figure 5 appears to show a small jump or gap around MES 1.0. This is an interesting artifact that warrants a brief comment in the Results or Discussion. Does this reflect a more pronounced visual difference between MES 0 and MES 1 in the training dataset, as alluded to in Section 5.4?

We now discuss this jump in the Limitations and Methodological Considerations section. This jump results from the dataset’s discrete class structure, which creates artificial boundaries and does not fully capture the continuous nature of biological disease progression. 

  1. Clarification of Interpretation of Figure 3: In Section 5.4, the authors state, "The jump in MES 0 shown in Fig. 3 indicates that..." However, Figure 3 displays performance metrics (FID, CMMD, etc.) across MES classes, not MES values themselves. The authors likely mean there is a significant change in the performance metrics between MES 0 and MES 1. It would be clearer to state this more precisely. For example, for AOE, the FID score improves (decreases) from ~42.5 at MES 0 to ~40.0 at MES 1 before rising again. This non-monotonic trend is worth discussing.

There was a typo in the figure referencing, we are now correctly referring to Figure 5 (ResNet-18 regression predictions on synthetically generated progression sequences using the Additive Ordinal Embedder with 0.01 increments between MES 0-3.).

As for the non-monotonic trend in the performance metrics, we added a new paragraph to discuss and present the results of our work more clearly, which also contributes to our conclusions.

“The observed non-monotonic performance trends across Mayo Endoscopic Score severity classes in Figure 3 further illuminate the relationship between dataset characteristics and generative model behavior. For less severe classes like MES 0 and 1 the model generates images with increased variety while preserving fidelity, resulting in improved FID scores but reduced precision. Conversely, high-severity classes like MES 2 and 3 suffer from limited training diversity due to their clinical rarity. The generated images for these classes exhibit greater diversity than the constrained training distribution, producing higher FID scores and lower precision while improving recall through enhanced coverage of underrepresented pathological manifestations. This pattern demonstrates that generative models can effectively augment rare disease classes by expanding beyond the limited diversity present in clinical datasets.”

  1. Strengthening a Claim with a Citation: In Section 4.1, the authors claim that "CLIP-based metrics provide more reliable assessment than Inception-based metrics for medical content evaluation." This is a strong statement. While plausible, it would be strengthened by citing a reference that supports this view or by briefly explaining the rationale  

Medical image datasets often have class imbalance, particularly with rarer cases. CLIP-based metrics are more robust to such an imbalance compared to traditional Inception-based metrics like FID. To clarify this point and support our choice of using CMMD, we have added new references that discuss these advantages in detail.

“Although FID scores are marginally higher for the ordinal methods, they perform better according to the CMMD metric. Literature reports that CMMD provides a more reliable assessment of medical content compared to Inception-based metrics [41,42].” 

Reviewer 2 Report

Comments and Suggestions for Authors

This paper proposes a method for progressive disease image generation based on the ordinal-aware diffusion models. The paper is clear and easy to follow; however, I recommend that the authors address the following comments:

  • To clarify the literature gap, the other embedding schemes should be discussed
  • It would be interesting to mention why diffusion models are used for your task
  • while the authors have provided some insights on the importance of including the AOE, for the sake of clarity, it would be better to dedicate a paragraph in Section 3 to explain that
  • Please discuss the computational complexity of the DM you proposed
  • To validate the proposed DM model, it would be better to compare it with some baseline generators or other SOTA methods (e.g., Lakas et al., Enhancing Diabetic Retinopathy Grading with Advanced Diffusion Models)  

Author Response

This paper proposes a method for progressive disease image generation based on the ordinal-aware diffusion models. The paper is clear and easy to follow; however, I recommend that the authors address the following comments:

To clarify the literature gap, the other embedding schemes should be discussed

We have added the following detail to our work to clarify the gap in the literature and highlight our contribution regarding novel ordinal embedding schemes:

“Non-ordinal embedding approaches treat all distinct classes as separate entities in a space. However, for medical datasets, it is imperative to employ ordinal embeddings particularly for the disease severity datasets which are inherently ordinal.”

It would be interesting to mention why diffusion models are used for your task

Recent advances in generative models, particularly diffusion models, present numerous research opportunities; accordingly, we have added a brief explanation of our motivation for using diffusion models to the introduction section.

“Our work combines the capabilities of state-of-the-art diffusion models and ordinal-aware embedding approaches and applies them to the gap in the literature for longitudinal data addressing the limitations of such approaches”

While the authors have provided some insights on the importance of including the AOE, for the sake of clarity, it would be better to dedicate a paragraph in Section 3 to explain that

We have added a new paragraph dedicated for ordinal embeddings and their benefits of using them.

“The AOE represents higher disease severity levels as incremental additions to the characteristics of lower levels, thereby capturing the progressive accumulation of pathological changes throughout disease progression. “

We’ve also expanded our previous explanation ("... the Additive Ordinal Embedder, which explicitly models the cumulative nature of pathological features.") as "The Additive Ordinal Embedder (AOE) explicitly models the cumulative nature of pathological features by representing higher disease severity levels as incremental additions to the characteristics of lower levels, thereby capturing the progressive accumulation of pathological changes throughout disease progression." for better clarity.

We have also added a new subsection “Complete Dataset Performance Assessment”  in the newly added Appendix B. This subsection provides a detailed comparison of  Basic Ordinal Embedder (BOE) and Additive Ordinal Embedder (AOE) performances using an equal number of synthetic images.

Please discuss the computational complexity of the DM you proposed

We have included an analysis of the computational complexity and parameter sizes of both proposed ordinal embedders to highlight their relatively low computational requirements.

“Compared to the standard CLIP text encoder with approximately 63 million parameters, both proposed ordinal embedders require only 3,072 parameters and maintain O(1) inference complexity, demonstrating significantly lower computational complexity.”

To validate the proposed DM model, it would be better to compare it with some baseline generators or other SOTA methods (e.g., Lakas et al., Enhancing Diabetic Retinopathy Grading with Advanced Diffusion Models)  

Our work addresses a distinct and novel problem setting of longitudinal disease progression sequence generation from image-based endoscopic data. This differs from typical image synthesis tasks focused on static image generation. To the best of our knowledge, there are no existing methods specifically designed to generate continuous progression models or intermediate disease stages. Consequently, direct comparison with prior diffusion-based or generative models that do not support longitudinal or ordinal progression synthesis is not feasible. Instead, we have focused our evaluation on ablations against baselines within our framework (i.e., CLIP) and validation of generated sequences through using regression predictions and UMAP manifold comparisons.

Reviewer 3 Report

Comments and Suggestions for Authors

The manuscript presents several notable contributions, including the development of ordinality-aware embedding strategies to effectively capture disease progression, the adaptation of advanced diffusion models with domain-specific modifications for medical image synthesis, and the generation of realistic synthetic longitudinal datasets enabling novel avenues for UC trajectory analysis. Furthermore, the integration of generative data augmentation techniques is explored to enhance deep learning model training with synthetic data. Considering these substantial contributions, I recommend the article for publication.

Author Response

The manuscript presents several notable contributions, including the development of ordinality-aware embedding strategies to effectively capture disease progression, the adaptation of advanced diffusion models with domain-specific modifications for medical image synthesis, and the generation of realistic synthetic longitudinal datasets enabling novel avenues for UC trajectory analysis. Furthermore, the integration of generative data augmentation techniques is explored to enhance deep learning model training with synthetic data. Considering these substantial contributions, I recommend the article for publication.

We thank the reviewer for their time and valuable feedback.